# The structure of the yeast Ctf3 complex

Stephen M Hinshaw[1]*, Andrew N Dates[2], Stephen C Harrison[1]*

[1]Department of Biological Chemistry and Molecular Pharmacology, Harvard Medical School, Howard Hughes Medical Institute, Boston, United States; [2]Harvard Chemical Biology PhD Program, Harvard University, Boston, United States

**Abstract** Kinetochores are the chromosomal attachment points for spindle microtubules. They are also signaling hubs that control major cell cycle transitions and coordinate chromosome folding. Most well-studied eukaryotes rely on a conserved set of factors, which are divided among two loosely-defined groups, for these functions. Outer kinetochore proteins contact microtubules or regulate this contact directly. Inner kinetochore proteins designate the kinetochore assembly site by recognizing a specialized nucleosome containing the H3 variant Cse4/CENP-A. We previously determined the structure, resolved by cryo-electron microscopy (cryo-EM), of the yeast Ctf19 complex (Ctf19c, homologous to the vertebrate CCAN), providing a high-resolution view of inner kinetochore architecture (Hinshaw and Harrison, 2019). We now extend these observations by reporting a near-atomic model of the Ctf3 complex, the outermost Ctf19c sub-assembly seen in our original cryo-EM density. The model is sufficiently well-determined by the new data to enable molecular interpretation of Ctf3 recruitment and function.
DOI: https://doi.org/10.7554/eLife.48215.001

## Introduction

Inner kinetochore proteins, particularly members of the Ctf19c, constitute the foundation of the kinetochore; they both recognize Cse4/CENP-A and recruit microtubule-interacting outer kinetochore proteins (*Musacchio and Desai, 2017*). Many of these factors were originally cloned and characterized for their contributions to faithful mitotic chromosome segregation (*Hinshaw and Harrison, 2018*). Specific Ctf19c activities that support chromosome segregation include specification of centromere-proximal origins as early-firing, recruitment of the Scc2/4 cohesin loading complex and its payload, the cohesin complex, and, as has been recently determined, recruitment of the chromosome passenger complex and its essential kinase subunit, Ipl1/Aurora B (*Fernius et al., 2013*; *García-Rodríguez et al., 2019*; *Halwachs et al., 2018*; *Natsume et al., 2013*). Recent findings also suggest a handoff between Ctf19c-dependent microtubule contact points, whereby the link between centromeric DNA and spindle microtubules shifts between chromosomal tethers during the course of the cell cycle (*Bock et al., 2012*; *Hara et al., 2018*). Understanding coordination of these diverse activities requires a structural picture of the inner kinetochore.

Both Scc2/4 recruitment and stable microtubule binding at later stages of the cell cycle depend on the conserved Ctf3/CENP-I complex (Ctf3c) (*Hara et al., 2018*; *Lang et al., 2018*; *Natsume et al., 2013*). The Ctf3c namesake, Ctf3, is a 733 amino-acid residue protein predicted to consist almost entirely of HEAT repeats (Huntingin, elongation factor 3 (EF3), protein phosphatase 2A (PP2A), and yeast kinase TOR1 repeats) (*Basilico et al., 2014*). Ctf3 associates with a coiled-coil dimer of Mcm16 and Mcm22 (Mcm16/22). The vertebrate Ctf3/CENP-I complex also includes the CENP-M protein, which has no known counterpart in yeast (*Basilico et al., 2014*). The Mcm16/22 heterodimer can be purified from recombinant sources, and the dimer is required for recovery of the full-length Ctf3 protein. Deletion of any one complex member perturbs recruitment of the other two to the kinetochore, and the intact complex binds and recruits the Cnn1-Wip1 dimer, which connects to microtubules through the four-protein Ndc80 complex (*Pekgöz Altunkaya et al., 2016*). In wild-

*For correspondence:
hinshaw@crystal.harvard.edu (SMH);
harrison@crystal.harvard.edu (SCH)

Competing interests: The authors declare that no competing interests exist.

type cells, kinetochore-localized Ctf3-GFP signal fluctuates throughout the cell cycle, appearing to peak in late anaphase before dropping as cells enter S phase (*Pot et al., 2003*).

A high-resolution Ctf3c structure, which would enable molecular interpretation of the functions listed above, has yet to be reported. Three-dimensional reconstructions of negatively-stained human CENP-I assemblies yielded an envelope for the complex consistent with the predicted HEAT repeat architecture for Ctf3/CENP-I, but the overall resolution was insufficient to support modeling beyond placement of putative domains (*Basilico et al., 2014*). Recent X-ray crystal structures showed the HEAT repeat architecture of an N-terminal fragment of fungal Ctf3 (Ctf3-N) bound with C-terminal fragments of Mcm16 and Mcm22 (Mcm16/22 C), providing a model for this part of the Ctf3c (*Hu et al., 2019*). Our previous cryo-EM reconstruction of the yeast Ctf19c enabled modeling of most of the ordered polypeptide chains, but the Ctf3c appeared flexibly attached to the core of the complex, limiting the apparent resolution for this region of the map and model (*Hinshaw and Harrison, 2019*). Consequently, we could not confidently assign a sequence register to the Ctf3 HEAT repeats we observed or to the Mcm16/22 coiled-coil. We now present a cryo-EM structure of the yeast Ctf3c, which resolves the overall organization of the complex, shows how Ctf3 binds Mcm16/22, and clarifies Ctf3c recruitment to the larger Ctf19c.

## Results

We purified the Ctf3c and determined its structure by single-particle cryo-EM (see Materials and methods). Initial images yielded two-dimensional averages, derived from a small hand-picked particle set, that matched projections of the density assigned to the Ctf3c in the full Ctf19c reconstruction (EMDB-0523). Additional blurry density was visible at the tips of several average particle images, likely indicating the presence of a domain (probably Ctf3-N and associated Mcm16/22 C, see below) without a fixed orientation relative to the well-resolved features. We then collected a large dataset. Particles recovered from these images yielded two-dimensional averages similar to those derived from screening images (*Figure 1C*). Particles contributing to two-dimensional averages that showed fine features were used for calculation of an initial three-dimensional

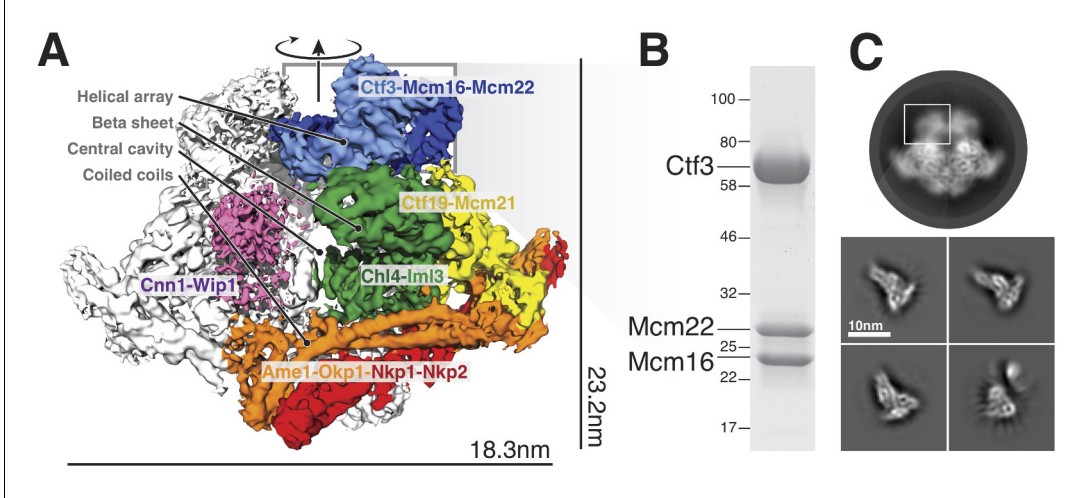

**Figure 1.** Overview of the Ctf19c and position of the Ctf3c within it. (**A**) Ctf19c density map colored according to constituent subcomplexes (*Hinshaw and Harrison, 2019*). (**B**) The recombinant Ctf3c sample used for cryo-EM (size markers – kDa). (**C**) Two-dimensional class averages showing the full Ctf19c (circle) and the Ctf3c (squares). The white box indicates the fuzzy Ctf19c density assigned to the Ctf3c.
DOI: https://doi.org/10.7554/eLife.48215.002
The following figure supplement is available for figure 1:

**Figure supplement 1.** Cryo-EM data processing The data processing procedure for Ctf3c structure determination is shown.
DOI: https://doi.org/10.7554/eLife.48215.003

reconstruction. Except for visual comparison of cryo-EM density, we did not include structural information from the previous Ctf19c reconstruction at any stage of model generation or refinement. The final map shows side-chain density throughout the particle, enabling modeling at near-atomic resolution (*Figure 1—figure supplement 1*).

The overall architecture of the Ctf3c agrees well with the features previously assigned to it in our Ctf19c reconstruction (*Hinshaw and Harrison, 2019*) (*Figure 2A*). Clearly-resolved side-chain density throughout the map allowed us to identify the amino-acid sequence of the Ctf3 domain visualized here and the corresponding fragments of Mcm16/22 (*Figure 2—figure supplements 1–2*, *Table 1*). We modeled five Ctf3-C HEAT repeats (*Figure 2B*), which have the overall fold of canonical HEAT arrays (*Perry and Kleckner, 2003*). Nearly 120 amino acid residues interrupt an imperfect sixth repeat at the C-terminal end of the modeled peptide chain, making a helical knob. As we saw in the previous Ctf3c model, an Mcm16/22 coiled-coil occupies the cleft formed by the concave surface of the HEAT array. The two main components of the structure, Ctf3-C and Mcm16/22 N, have opposite polarities; the central parts of Mcm16 and Mcm22 form a parallel coiled-coil that begins near the C-terminal end of the modeled Ctf3 polypeptide. The N-terminal parts of Mcm16 and Mcm22, form a set of four interdigitated helices that pack onto the convex side of the C-terminal part of the HEAT array and contact the helical knob.

Sequence conservation analysis suggested structural similarity between the N- and C-terminal parts of human CENP-I (*Basilico et al., 2014*). Indeed, Ctf3/CENP-I has a tandem HEAT repeat architecture, the N-terminal part of which was resolved crystallographically (PDB 3Z07) (*Hu et al., 2019*), and the C-terminal part we have resolved here. Our cryo-EM densities for Ctf3 (EMDB-0523 and here) imply that the two sets of Ctf3 HEAT repeats do not have a fixed relative orientation. Nevertheless, only 7 and 12 amino acid residues for Mcm16 and Mcm22 (respectively) connect the ordered regions that coordinate the two Ctf3 globular domains (*Hu et al., 2019*) (*Figure 2C*). The final model shows that the structure determined here (Ctf3-C-Mcm16/22 N) is essentially the precise complement of the Ctf3-N-Mcm16/22 C crystal structure (*Hu et al., 2019*). Therefore, the Mcm16/22 dimer extends without a major interruption along the Ctf3 HEAT repeats, forming two tightly but flexibly linked structural modules: Ctf3-N-Mcm16/22 C and Ctf3-C-Mcm16/22 N.

Ctf3c recruitment to the kinetochore depends on interactions with other Ctf19c proteins (*Measday et al., 2002*). We previously found that Ctf3c recruitment depends on amino acid residues 1–95 of Mcm21 (Mcm21-N), a region of the peptide chain not visible in published crystal structures (*Schmitzberger and Harrison, 2012*; *Schmitzberger et al., 2017*). The high-resolution Ctf3c map we report here allowed us to identify features in the previous Ctf19c cryo-EM map that are not visible in the current model and therefore must correspond to Ctf19c proteins not included in the current reconstruction. First, a small helical fragment packs into the cleft between the convex surface of the Ctf3 HEAT array and the Mcm16/22 N-terminal helices in the Ctf19c map (*Figure 3A*). This fragment likely accommodates a presumed short helical motif in Mcm21-N (Mcm21-10-34) (*García-Rodríguez et al., 2019*; *Hinshaw and Harrison, 2019*). Second, the Ctf3c map lacks density for an extended peptide that snakes along the solvent-exposed surface of the Mcm16/22 N-terminal helices. This extended peptide emanates from the first resolved Mcm21 residue (at its N-terminus; Mcm21-Ser153) and likely connects through a loop not visualized in the Ctf19c map to the short helix mentioned above.

The new Ctf3c map also enabled unambiguous identification of the Ctf3 residues that contact Iml3 and likely contribute to Ctf3c kinetochore recruitment. Docking the new Ctf3c model determined at high resolution into the Ctf19c map shows that density for bulky Ctf3 residues can be accounted for by the new model (*Figure 3B*). Continuous density connecting Ctf3 and Iml3 accommodates Ctf3-Arg366, which packs onto Iml3-Trp112, likely enabling hydrophobic contact between the tryptophan indole and the aliphatic part of the arginine side chain. The neighboring Ctf3-Lys365 is well-positioned to make a salt bridge with Iml3-Glu82 in the full Ctf19c. To test whether this interface, like the one we previously reported between Ctf3 and Mcm21-N, contributes to Ctf3c kinetochore recruitment, we mutated three protruding residues to create the *ctf3-SDD* allele (Ctf3-W362S, K365D,R366D). Yeast cells expressing Mcm22-GFP (to mark Ctf3c localization), Spc110-mCherry (to mark the spindle pole body/kinetochore), and Ctf3-WT display Mcm22-GFP localization that matches that of Ctf3-GFP (*Hinshaw and Harrison, 2019*). Cells lacking Iml3 (*iml3Δ*) or expressing the *ctf3-SDD* allele showed defective Mcm22-GFP localization, a phenotype that was less severe in *ctf3-SDD* cells than in *iml3Δ* cells but was reproducible in three independently-derived *ctf3-SDD* strains

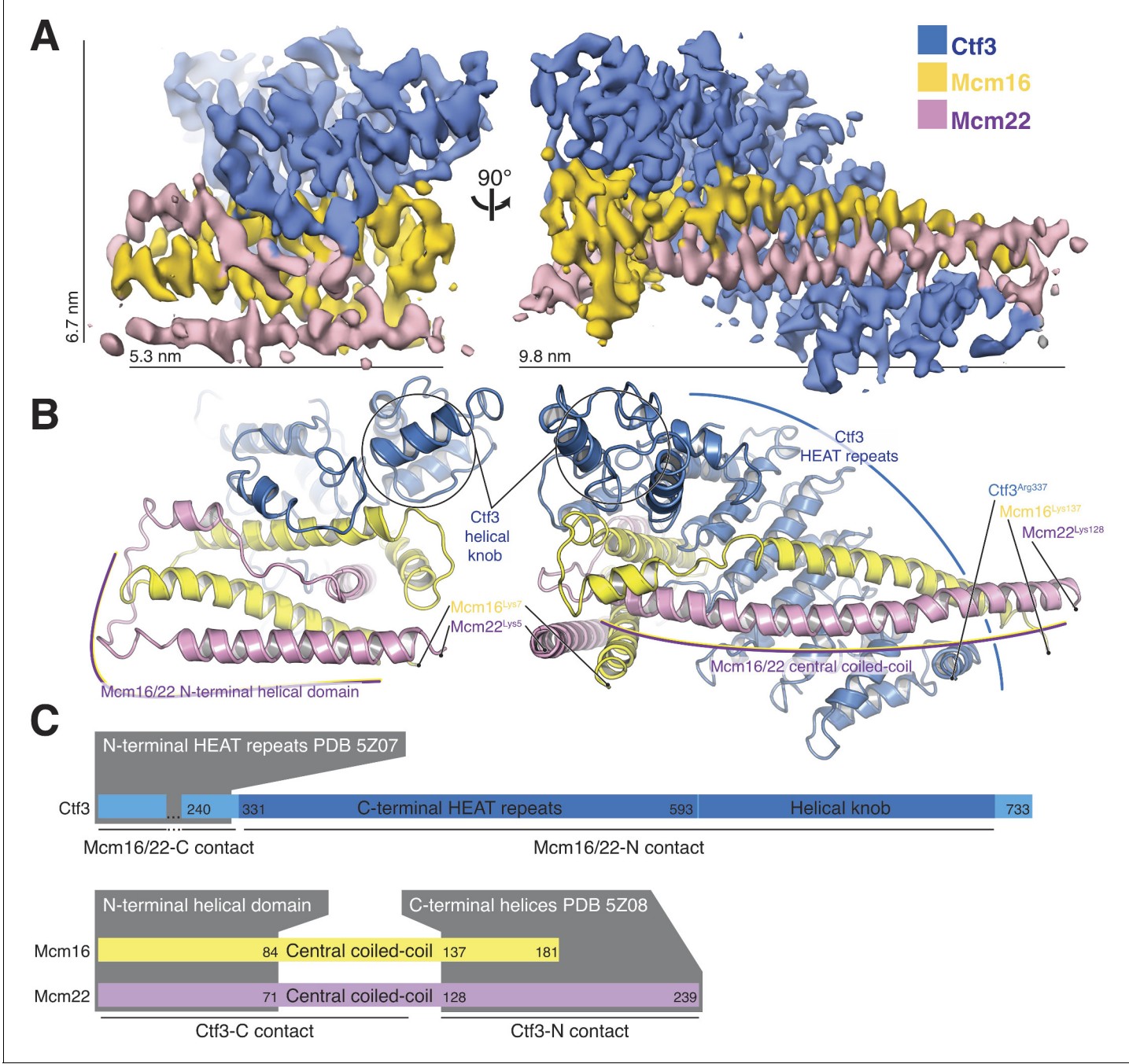

**Figure 2.** Overview of the Ctf3c structure determined by cryo-EM. (**A**) Cryo-EM density map shown in in two orientations. The density is colored according to the atomic model of the complex. (**B**) Atomic model of the Ctf3c colored according to panel (**A**) and shown in the same orientations. Prominent Ctf3c features are labeled. (**C**) Domain diagram depicting Ctf3c components. Lines below the colored bars indicate Ctf3-Mcm16/22 contacts. Text inside the colored bars and gray boxes indicates domain structure. PDB numbers refer to the published crystal structures of Ctf3-N and Mcm16/22 C (*Hu et al., 2019*).

DOI: https://doi.org/10.7554/eLife.48215.004

The following figure supplements are available for figure 2:

**Figure supplement 1.** Analysis of cryo-EM density map quality and model fitting.

DOI: https://doi.org/10.7554/eLife.48215.005

**Figure supplement 2.** Fourier shell correlation curves describing the Ctf3c density map (half-map to half-map) and map-to-model correlations.

DOI: https://doi.org/10.7554/eLife.48215.006

**Table 1.** Model information.

| Protein | Chain ID | HOMOLOGS (Human; pombe) | Chain length (Total; Modeled) | Template | Procedure | Modelled residues |
|---------|----------|--------------------------|-------------------------------|----------|-----------|-------------------|
| Mcm16 | H | CENP-H; Fta3 | 181; 130 | – | Build de novo | 7–137 |
| Ctf3 | I | CENP-I; Mis6 | 733; 381 | 6NUW | Modify, build de novo | 337–718 |
| Mcm22 | K | CENP-K; Sim4 | 239; 124 | – | Build de novo | 5–128 |

DOI: https://doi.org/10.7554/eLife.48215.011

(*Figure 3C*, *Figure 3—figure supplement 1*). Therefore, the Iml3-Ctf3 interface we observe in cryo-EM reconstructions of the full Ctf19c recapitulates the one used by cells to recruit the Ctf3c to the kinetochore.

## Discussion

There are three published Ctf3c/CENP-I complex models (*Basilico et al., 2014*; *Hinshaw and Harrison, 2019*; *Hu et al., 2019*). The volume derived from a negative-stain electron microscopy reconstruction of the four-protein human CENP-I complex (CENP-H/I/K/M) could likely accommodate the model we present here, along with the N-terminal part of the Ctf3/CENP-I and human CENP-M. Our recent Ctf3c structure, which we determined in the context of the full Ctf19c, provides information about the mechanism of Ctf3c recruitment but did not allow us to confidently assign amino acid

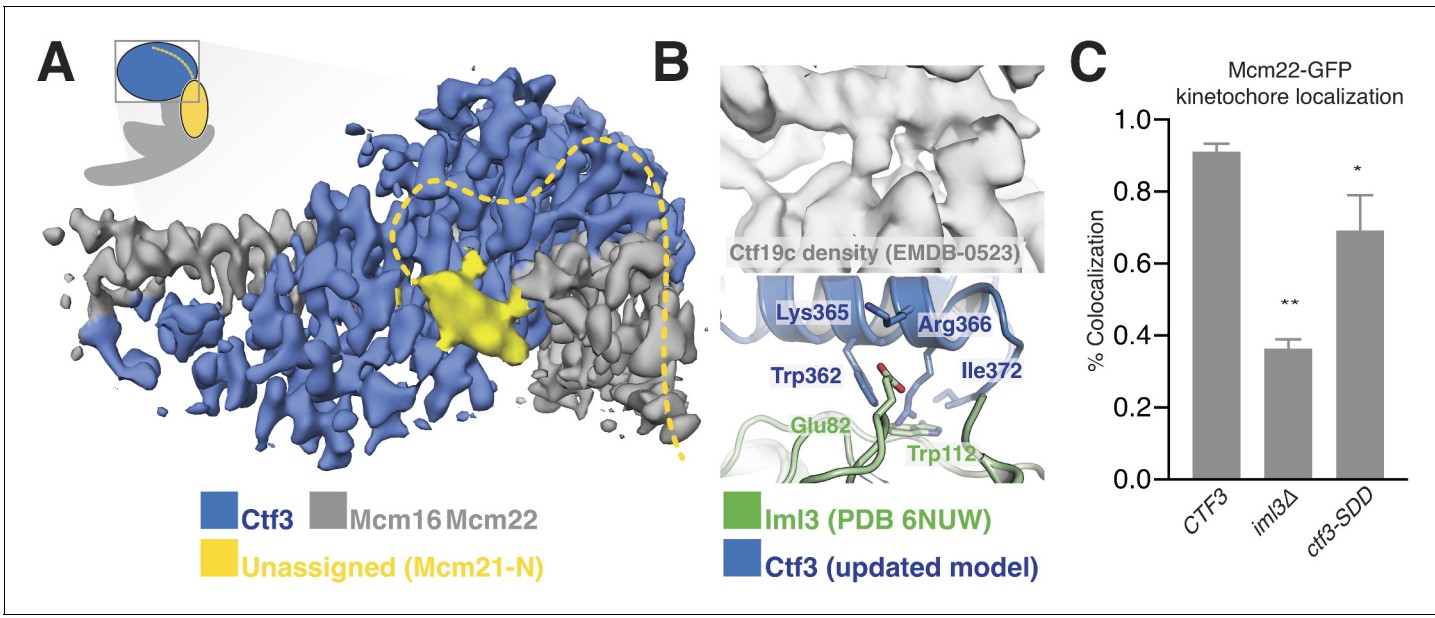

**Figure 3.** Interactions between the Ctf3c and the full Ctf19c. (**A**) Composite cryo-EM density map showing extra density, only visible in the original Ctf19c map, which likely belongs to Mcm21-N. The image was created by combining the high-resolution Ctf3c map (blue and gray) with the displayed yellow density from the Ctf19c map (EMDB-0523). The inset shows the position of the map in the context of the full Ctf19c model. (**B**) Cryo-EM density showing the interaction between Ctf3 and Iml3. The top panel shows density determined for the interface in the context of the full Ctf19c. The bottom panel shows a composite model; Iml3 coordinates refined into the Ctf19c cryo-EM map are green, and Ctf3 coordinates refined into the current density map are blue. (**C**) The indicated *S. cerevisiae* strains (*CTF3 – CTF3-3FLAG*: SMH690, SMH691, SMH692; *iml3Δ*: SMH91; *ctf3-SDD – ctf3-SDD-3FLAG*: SMH693, SMH694, SMH695) expressing Mcm22-GFP and Spc110-mCherry were imaged during asynchronous division. Coincidence frequencies of Mcm22-GFP foci with Spc110-mCherry foci are plotted (error bars –±SD for frequency measurements from three independent cultures; * – $p<0.05$; ** – $p<0.01$, Student's t-test, two tails, unequal variance).

DOI: https://doi.org/10.7554/eLife.48215.007

The following figure supplement is available for figure 3:

**Figure supplement 1.** Expression levels of Ctf3 WT and mutant proteins.

DOI: https://doi.org/10.7554/eLife.48215.008

sequence to the density. Finally, a recent crystal structure of Ctf3 from *Chaetomium thermophilum* in complex with Mcm16/22 from a related species (*Thielavia terrestri*) provides a high-resolution view of the Ctf3/CENP-I N-terminal domain and of the Mcm16/22 C-terminal regions but no information about the domains we report here.

Incorporation of the updated Ctf3c model into our previous model of the Ctf19c yields a nearly complete structural description of this 13-protein kinetochore anchor (*Figure 4A*). Newly-resolved interfaces between Ctf3 and the Ctf19c proteins Iml3 and Mcm21 enable direct visualization and, consequently, modulation of Ctf3c recruitment. A more precise understanding of Ctf3c recruitment will need to account for dynamic localization of Ctf3 and its binding partner Cnn1 during the cell cycle. Post-translational modification of Ctf3c proteins, Cnn1, Mcm21, Mif2/CENP-C, or some combination of these factors may explain this behavior.

A major open question, brought into focus by the current cryo-EM reconstruction of the Ctf3c, is whether and in what way the N-terminal domain of Ctf3 contacts the Cse4 nucleosome. Although Ctf3-N was not clearly visible in our cryo-EM reconstruction of the Ctf19c, we did see poorly-resolved density at the N-terminal tip of the Ctf3 HEAT repeat array, which we assigned to the Cnn1-Wip1 dimer. Reanalysis of this density map suggests Ctf3-N also contributes to the observed density, an arrangement that agrees with the identification of contact between Ctf3c and Cnn1-Wip1 by crosslinking-coupled mass spectrometry (*Pekgöz Altunkaya et al., 2016*). If so, then the entire Ctf3N-Cnn1-Wip1 module must swing outward (away from the Ctf19c central cavity) to make way for the Cse4 nucleosome, flanking DNA, and associated CBF3 complex factors (*Figure 4B*). This

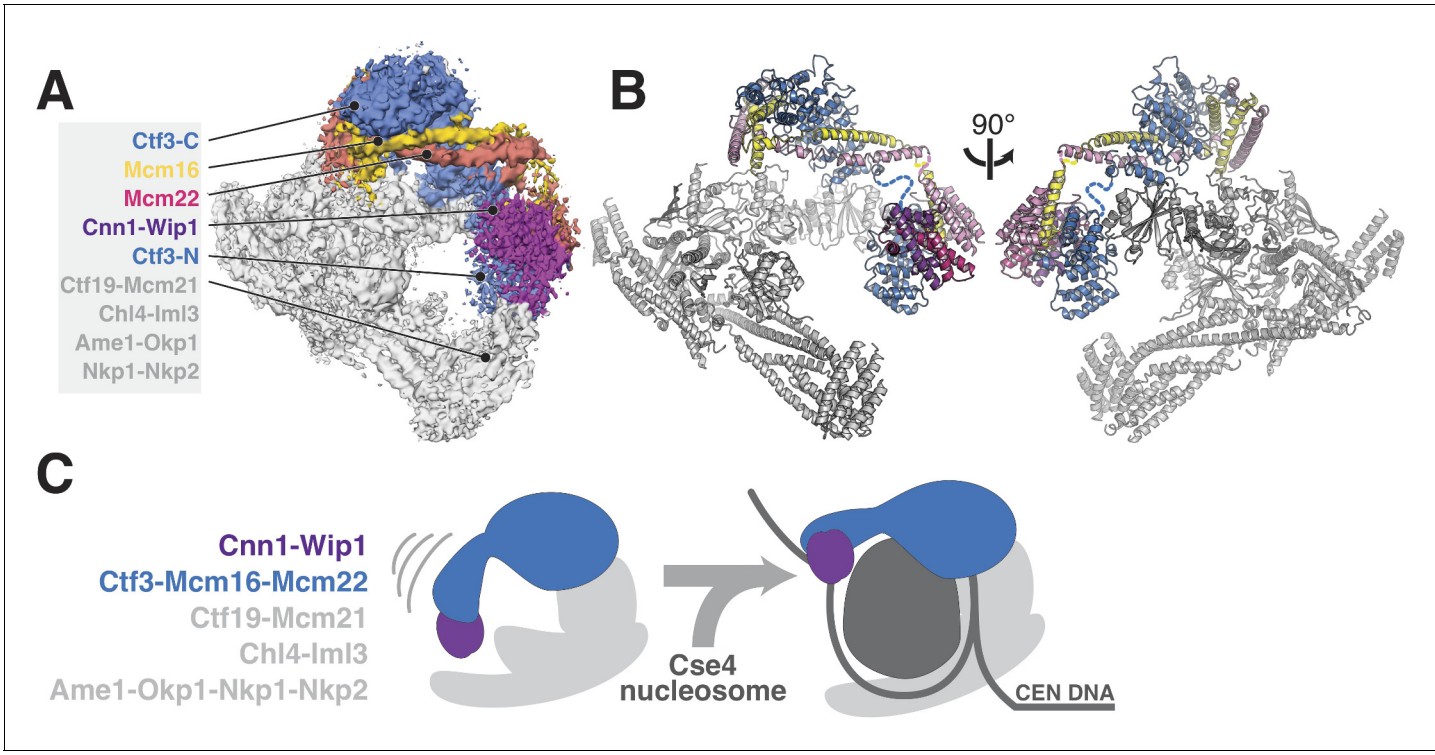

**Figure 4.** Updated model of Ctf19c architecture and interactions. (A–B) Cryo-EM density map (A) and corresponding composite atomic model (B) showing the 13-protein Ctf19c in its monomeric form (EMDB-0523). The Ctf3 N-terminal HEAT repeat domain and adjacent Cnn1-Wip1 heterodimer, both of which occupy updated positions in the overall model, are labeled. The Ctf3c components visualized in the current study are also labeled. The Ctf3-N-Mcm16/22 C module (PDB 5Z08), with Ctf3-N substituted by a homology model, was positioned to minimize the gap between ordered parts of Mcm16/22 (reported here and previously) (*Hu et al., 2019*). The interaction between Ctf3-N and Cnn1-Wip1 has not been visualized experimentally, so the histone-fold extension of Cnn1 required for Ctf3 contact (*Pekgöz Altunkaya et al., 2016*) was used to position Cnn1-Wip1 with respect to Ctf3-N. (C) Proposed model for nucleosome accommodation by the Ctf19c in its monomeric form. Ctf3-N, along with Cnn1-Wip1, flex up and outward to make space for a Cse4/CENP-A nucleosome. Current data does not distinguish between one or two copies of the Ctf19c decorating the Cse4/CENP-A particle. A single Ctf19c was therefore drawn for clarity.

DOI: https://doi.org/10.7554/eLife.48215.009

outward displacement would position Cnn1-Wip1 to contact and stabilize nucleosome-flanking DNA, a suggestion consistent with the observation that the vertebrate CENP-T/W heterodimer binds and bends DNA (*Nishino et al., 2012*; *Takeuchi et al., 2014*). How this reorganization works and how cell cycle-dependent enzymatic activities regulate it are matters of ongoing investigation.

# Materials and methods

## Key resources table

| Reagent type (species) or resource | Designation | Source or reference | Identifiers | Additional information |
|---|---|---|---|---|
| Gene (*S. cerevisiae*) | See *Table 2* | | | |
| Strain, strain background (*S. cerevisiae*) | S288c | | | |
| Genetic reagent (*S. cerevisiae*) | See *Table 2* | | | |
| Antibody | anti-FLAG-HRP (mouse monoclonal) | Sigma | A8592 | (1:1000) |
| Antibody | anti-PGK1 (mouse monoclonal) | Invitrogen | 459250 | (1:5000) |
| Antibody | goat anti-mouse-IgG-HRP (rabbit polyclonal) | Abcam | Ab97046 | (1:10000) |
| Recombinant DNA reagent | See *Table 3* | | | |
| Cell line (*E. coli*) | Rosetta 2(DE3)pLysS; *E. coli* | EMD Millipore | 71403 | Harrison lab stock |
| Software, algorithm | SerialEM (v3.7) | *Mastronarde, 2005* | | |
| Software, algorithm | MotionCor2 (v1.1.0) | *Zheng et al., 2017* | | |
| Software, algorithm | CTFFIND4 (v4.1.8) | *Rohou and Grigorieff, 2015* | | |
| Software, algorithm | Relion (v3.0.1) | *Kimanius et al., 2016* | | |
| Software, algorithm | ResMap (v1.1.4) | *Kucukelbir et al., 2014* | | |
| Software, algorithm | PyMol (v2.1.0) | Schrödinger, LLC | | |
| Software, algorithm | Chimera (v1.11.2) | *Pettersen et al., 2004* | | |
| Software, algorithm | Coot (v0.8.8) | *Emsley et al., 2010* | | |
| Software, algorithm | Phenix (v1.13) | *Afonine et al., 2018* | | |
| Software, algorithm | TrackMate (v3.0.0) | *Tinevez et al., 2017* | | |
| Software, algorithm | MAFFT | *Katoh et al., 2017* | | |
| Software, algorithm | JalView | *Waterhouse et al., 2018* | | |
| Software, algorithm | Phyre2 | *Kelley et al., 2015* | | |
| Software, algorithm | Fiji | *Schmitzberger and Harrison, 2012* | | |
| Software, algorithm | python 2.7.2 | www.python.org | | |
| Other (holey carbon grids) | C-flat | Electron Microscopy Sciences | CF-2/1–4C | |

## Protein purification and Cryo-EM sample preparation

*S. cerevisiae* Ctf3c was expressed and purified as described previously (*Hinshaw and Harrison, 2019*; *Hinshaw et al., 2017*). The three protein-coding genes were expressed as a single transcript with three ribosome binding sites. After recombinant protein expression in *E. coli*, cell pellets were frozen and stored at −80°C in lysis buffer until use. Purified protein samples (~800 μg from 24 L of bacterial growth) were supplemented with 5% glycerol by volume and stored at ~8 mg/mL at −80°C. For cryo-EM grid preparation, an aliquot of frozen Ctf3c stock was thawed and diluted to ~1.8 mg/

**Table 2.** Yeast strains used in this study.

| Strain number | Genotype | Reference |
|---|---|---|
| SMH81 | MATa MCM22-GFP::HisMX SPC110-mCherry::hphMX | (Huh et al., 2003) |
| SMH91 | MATa MCM22-GFP::HisMX SPC110-mCherry::hphMX iml3Δ::KanMX | This study |
| SMH690 | MATa MCM22-GFP::HisMX SPC110-mCherry::hphMX CTF3-3FLAG::KanMX | This study |
| SMH691 | MATa MCM22-GFP::HisMX SPC110-mCherry::hphMX CTF3-3FLAG::KanMX | This study |
| SMH692 | MATa MCM22-GFP::HisMX SPC110-mCherry::hphMX CTF3-3FLAG::KanMX | This study |
| SMH693 | MATa MCM22-GFP::HisMX SPC110-mCherry::hphMX ctf3-SDD-3FLAG::KanMX | This study |
| SMH694 | MATa MCM22-GFP::HisMX SPC110-mCherry::hphMX ctf3-SDD-3FLAG::KanMX | This study |
| SMH695 | MATa MCM22-GFP::HisMX SPC110-mCherry::hphMX ctf3-SDD-3FLAG::KanMX | This study |

DOI: https://doi.org/10.7554/eLife.48215.012

mL in gel filtration buffer (150 mM NaCl, 20 mM tris-HCl pH 8.5, 1 mM TCEP). The diluted sample was applied to C-flat holey carbon grids (Electron Microscopy Sciences, CF-1.2/1.3–3C for screening, CF-2/1–4C for data collection) and plunge-frozen in liquid ethane on a Cryo-Plunge three instrument (Gatan). Grids were prepared at ~90% relative humidity. 3.5 µL of sample was applied to negatively-discharged grids, which were subsequently blotted for 4 s from both sides with minimal delay.

## Cryo-EM data collection

A screening dataset was collected manually on an F20 instrument (FEI) using the UCSF *Figure 4* semi-automated data collection software package (*Li et al., 2015*). A single large dataset was collected on a Titan Krios G3i (FEI). The instrument was outfitted with a pre-camera energy filter (Gatan Image Filter) and operated at 300 kV. Images were collected in nanoprobe mode at spot size three with an illuminated area of ~1 µm at the sample level. Images were collected on a K3 camera (Gatan) in counting mode at a nominal magnification of 105 kx with a pixel size of 0.85 Å$^2$ (*Table 4*). Dose-fractionated movies were collected (3 s per movie, 50 frames, 17 electrons per Å$^2$ per second, and 1.0 electrons per Å$^2$ per frame). Five movies were collected for each hole on the grid, and nine holes were imaged at each stage position, yielding 45 movies per round of stage movement. We used SerialEM v3.7 with on-the-fly beam tilt correction of image shift-induced coma to collect the high-resolution dataset (*Mastronarde, 2005*). Computational beam tilt correction during data processing further improved the experimental density (*Figure 1—figure supplement 1*).

## Cryo-EM data processing

2-dimensional class average particle images were first generated from ~1500 hand-picked particles from the screening dataset (F20). Class averages judged to represent distinct views of the Ctf3c were then used as references for automated particle picking against the full screening dataset. The resulting particles were further classified, and 11 of the best class averages were used as particle picking references for the high-resolution dataset.

For high-resolution data processing, sample displacements and defocus estimates were calculated during data collection using Motioncor2 v1.1.0 (5-by-5 patch correction) and CTFFIND4 v4.1.8 (*Rohou and Grigorieff, 2015*; *Zheng et al., 2017*). All data processing steps were carried out in RELION-3.0 unless otherwise noted (*Zivanov et al., 2018*) (*Figure 1—figure supplement 1*). Corrected movies with maximum resolution estimations (generated by CTFFIND4) greater than 4.75 Å were discarded. Particles were picked using the class average images described above, and the

**Table 3.** Plasmids used in this study.

| Plasmid number | Genotype | Reference |
|---|---|---|
| pSMH145 | pLIC-Tra His6-TEV-Ctf3; His6-TEV-Mcm16; His6-TEV-Mcm22 | (Hinshaw et al., 2017) |
| pSMH1269 | pFA6a-CTF3-6Gly-3FLAG-KanMX6 (BsiWI/SalI) | This study |
| pSMH1577 | pFA6a-ctf3-SDD-6Gly-3FLAG-KanMX6 (BsiWI/SalI) | This study |

DOI: https://doi.org/10.7554/eLife.48215.013

**Table 4.** Cryo-EM data collection, refinement, and validation.

| | CTF3c (EMDB-20200) (PDB 6OUA) |
|---|---|
| **Data collection and processing** | |
| Magnification | 105,000 |
| Voltage (kV) | 300 |
| Electron exposure (e–/Å$^2$) | 51 |
| Defocus range (μm) | −1.5 to −3.5 |
| Pixel size (Å) | .85 |
| Symmetry imposed | None |
| Initial particle images (no.) | 2,062,207 |
| Final particle images (no.) | 71,632 |
| Map resolution (Å) FSC threshold | 4.18 (0.143) |
| | |
| **Refinement** | |
| Initial model used (PDB code) | 6NUW |
| Model resolution (Å) FSC threshold | 4.24 (0.5) |
| Model resolution range (Å) | 40–4.18 |
| Map sharpening $B$ factor (Å$^2$) | −151.424 |
| Model composition Non-hydrogen atoms Protein residues Ligands | 5208 637 0 |
| $B$ factors (Å$^2$) | Min: 36.02 Max: 141.08 Mean: 72.17 |
| R.m.s. deviations Bond lengths (Å) Bond angles (°) | 0.008 1.351 |
| Validation MolProbity score Clashscore Poor rotamers (%) | 2.37 7.09 2.35 |
| Ramachandran plot Favored (%) Allowed (%) Disallowed (%) | 82.25 99.68 0.32 |

DOI: https://doi.org/10.7554/eLife.48215.010

resulting particle stack was subjected to two-dimensional classification. Following previously-described methods for reconstructing small particles, we performed a second round of two-dimensional classification to select and enrich for underrepresented projections (projections down the long aspect of the particle) (*Herzik et al., 2019*). Subsequent processing steps are described in *Figure 1—figure supplement 1*. We noticed large particle movements with apparently poor local correlation in our movie files, leading to major improvements in map quality upon per-particle motion correction (particle polishing steps).

## Model building, Refinement, and Analysis

We used as a starting point for model building the original Ctf3c structure (PDB 6NUW), which was predominantly composed of poly-alanine chains. Manual model adjustments were done in Coot v0.8.8 (*Emsley et al., 2010*). We used secondary structure predictions generated by Phyre2 to guide modeling (*Kelley et al., 2015*). Helical fragments corresponding to HEAT repeats 1–5 fit the density

well, and the higher-resolution map enabled de novo modeling of the Ctf3 helical knob. The Mcm16/22 coiled-coils were originally placed in the reverse orientation. Inversion of the corresponding sequence resulted in a good fit of the density throughout the chains, including the N-terminal helices external to the Ctf3 cavity. The rebuilt and manually adjusted model was subjected to real-space refinement in Phenix v1.13 with global minimization, atomic displacement parameters (ADP), and local grid searches (*Afonine et al., 2018*). We used the phenix.secondary_structure_restrains program to generate restraints for modeled alpha-helices, which we then edited manually. In later rounds of coordinate refinement, we also used simulated annealing.

The Ctf3c model was fit into the Ctf19c density map in Chimera using the Fit in Map command (*Pettersen et al., 2004*). A homology model of *S. cerevisiae* Ctf3-N was generated using Swiss Model with PDB 3Z07 as a template (*Hu et al., 2019*; *Waterhouse et al., 2018*). A crystal structure of the histone fold domain of chicken CENP-T/W (PDB 3B0C) was docked into the density adjacent to the Ctf3-N model (*Nishino et al., 2012*). The ResMap program, implemented in Relion-3.0, was used to determine local resolution (*Figure 2—figure supplement 1A*) (*Kucukelbir et al., 2014*).

### Yeast strain construction, imaging, and Western blot

Yeast strains were created by integration of PCR products coding for 3-FLAG-tagged Ctf3 or Ctf3-SDD according to standard methods (*Longtine et al., 1998*). For live-cell imaging (*Figure 3C*), three independent clones each were examined for the *CTF3*-3FLAG and *ctf3-SDD*-3-FLAG genotypes. Three cultures were imaged for each genotype. Fluorescence imaging and image processing were carried out exactly as described previously (*Hinshaw and Harrison, 2019*). Cells were grown in synthetic complete medium until mid-log phase and diluted into fresh medium ~2 hr before imaging on a Nikon Ti2 fluorescence microscope with Perfect Focus System and a Nikon Plan Apo 60 × 1.4 NA oil-immersion objective lens. The microscope was outfitted with a Tokai Hit stage-top incubator and a Hamamatsu Flash 4.0 V2 +sCMOS camera. Imaging and image segmentation protocols and settings were identical to those we described previously (*Hinshaw and Harrison, 2019*; *Tinevez et al., 2017*). At least two stage positions and data from at least 1,000 mCherry foci (fiducials used for the Mcm22-GFP focus calling routine) were averaged for each replicate. Western blotting (*Figure 3—figure supplement 1*) was performed as described previously (*Hinshaw and Harrison, 2019*).

## Acknowledgements

We thank the staff at the Harvard Cryo-Electron Microscopy Center for Structural Biology for help with high-resolution cryo-EM data collection, particularly Richard Walsh, Sarah Sterling, and Zongli Li. We thank Shaun Rawson for real-time movie correction and for helpful comments during data processing. We thank SBGrid for computational support, particularly Mick Timony and Justin O'Connor. We thank Jennifer Waters and the staff at the Nikon Imaging Center at Harvard Medical School for light microscopy support. We thank Simon Jenni for guidance in implementing the HEALPix library and for helpful discussions. SMH is an HHMI fellow of the Helen Hay Whitney Foundation. DND was supported by the Harvard Chemical Biology Graduate Program (National Institutes of Health 5T32GM095450-08). SCH is an Investigator of the Howard Hughes Medical Institute.

## Additional information

### Funding

| Funder | Grant reference number | Author |
|---|---|---|
| Howard Hughes Medical Institute | | Stephen M Hinshaw<br>Stephen C Harrison |
| Helen Hay Whitney Foundation | | Stephen M Hinshaw |
| National Institutes of Health | 5T32GM095450-08 | Andrew N Dates |

The funders had no role in study design, data collection and interpretation, or the decision to submit the work for publication.

## Author contributions
Stephen M Hinshaw, Conceptualization, Funding acquisition, Investigation, Methodology, Writing—original draft, Writing—review and editing; Andrew N Dates, Data curation, Investigation; Stephen C Harrison, Supervision, Funding acquisition, Writing—review and editing

## Author ORCIDs
Stephen M Hinshaw ⅈD https://orcid.org/0000-0003-4215-5206
Stephen C Harrison ⅈD https://orcid.org/0000-0001-7215-9393

## Decision letter and Author response
Decision letter https://doi.org/10.7554/eLife.48215.021
Author response https://doi.org/10.7554/eLife.48215.022

## Additional files

### Supplementary files
• Transparent reporting form
DOI: https://doi.org/10.7554/eLife.48215.014

### Data availability
The final Ctf3c cryo-EM map and the soft mask used for three-dimensional refinements of fully unbinned particles have been deposited with accession code EMD-20200. The refined Ctf3c model has been deposited with accession code 6OUA.

The following previously published datasets were used:

| Author(s) | Year | Dataset title | Dataset URL | Database and Identifier |
|---|---|---|---|---|
| Hinshaw SM, Harrison SC | 2019 | Yeast Ctf19 complex | http://www.ebi.ac.uk/pdbe/entry/emdb/EMD-0523 | EMBL-EBI, EMD-0523 |
| Hinshaw SM, Harrison SC | 2019 | Cryo-EM structure of the yeast Ctf3 complex | https://www.rcsb.org/structure/6OUA | Protein Data Bank, 6OUA |

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
