## [Decision Letter]

Thank you for submitting your article "The structure of the yeast Ctf3 complex" for consideration by *eLife*. Your article has been reviewed by two peer reviewers, including Andrea Musacchio as the Reviewing Editor and Reviewer #1, and the evaluation has been overseen Cynthia Wolberger as the Senior Editor. The following individual involved in review of your submission has also agreed to reveal his identity: Hongtao Yu (Reviewer #2).

The reviewers have discussed the reviews with one another and the Reviewing Editor has drafted this decision to help you prepare a revised submission.

Summary:

This concise contribution reports a new EM structure that allows a more accurate description and understanding of the Ctf19 complex. Specifically, the authors studied a fragment of Ctf3, Mcm16, and Mcm22 roughly encompassing half of the Ctf3 complex (the N-terminal halves of Mcm16 and Mcm22, and the C-terminal half of Ctf3). At the resolution of the map, the authors were finally able to complete a univocal assignment of sequence to density. The resulting model supersedes, in various ways, the original assignments for this sub-complex initially released with a recent *eLife* paper by the same authors (including reversing the direction of the Mcm16 and Mcm22 helices). The chosen Research Advance format seems ideal for the manuscript's content. The authors should only consider the following minor points:

Minor points:

1) Figure 2C: It is somewhat confusing, in particular the grey shapes. What is meant by 'N-terminal helical domain'? Aren't the N-terminal regions of Mcm16-Mcm22 in contact with the C-terminal region of Ctf3? If this information is in the figure, it does not come across very straightforwardly.

2) Results: 'identification the Ctf3' ('of' missing)

3) "nucleosome-flaking DNA" should be "nucleosome-flanking DNA".

4) In Figure 4C, please indicate that the dark gray lines represent DNA.

---

## [Author Response]

Minor points:1) Figure 2C: It is somewhat confusing, in particular the grey shapes. What is meant by 'N-terminal helical domain'? Aren't the N-terminal regions of Mcm16-Mcm22 in contact with the C-terminal region of Ctf3? If this information is in the figure, it does not come across very straightforwardly.

We have updated Figure 2C to show Ctf3-Mcm16/22 contacts below the colored bars. The gray boxes now just refer to the overall domain architecture and, where appropriate, the corresponding PDB number. We have added a short note to the legend to clarify.

2) Results: 'identification the Ctf3' ('of' missing)

We fixed this.

3) "nucleosome-flaking DNA" should be "nucleosome-flanking DNA".

We fixed this.

4) In Figure 4C, please indicate that the dark gray lines represent DNA.

We added “CEN DNA” to the figure above the corresponding line.